# Initial Analysis of the Effectiveness of Compass-Behavioral for Autistic Youth: A Community-Based Retrospective Analysis

**DOI:** 10.3390/bs15121667

**Published:** 2025-12-03

**Authors:** Doreen Samelson, Jennifer Ikola, Brianna Fitchett, Michelle Befi, Vincent Bemmel, Ben Pfingston, Lindsey Sneed

**Affiliations:** 1Catalight Research Institute, 2730 Shadelands Dr., Walnut Creek, CA 94598, USA; 2Easterseals Northern California, 2730 Shadelands Dr. Building 10, Walnut Creek, CA 94598, USA

**Keywords:** autism, neuroaffirming, coaching, executive functioning, community-based

## Abstract

Compass-Behavioral is a novel, structured coaching program designed to enhance executive functioning and wellbeing in autistic youth and adults. This mixed-methods study examined outcomes from program participation across multiple clinical sites within a nonprofit organization. Quantitative analyses used de-identified archival data from participants completing the Youth and Adult Wellbeing Scales, along with self-assessments associated with the program. Results demonstrated trends toward improvement in wellbeing following program participation. Results of the self-assessment improved significantly from baseline to completion. Complementary qualitative data were gathered from participant feedback surveys, providing insights into perceived benefits and areas for program improvement. Themes included increased self-awareness, improved organization and planning, and greater confidence in everyday functioning. Taken together, the findings suggest Compass-Behavioral is a promising program for supporting executive functioning and quality of life outcomes among autistic individuals, while also pointing to opportunities for strengthening measurement and long-term impact. Further research is warranted to refine outcome measures, expand sample sizes, and evaluate program effects in more diverse clinical and community contexts.

## 1. Introduction

People with Autism Spectrum Disorder (ASD; hereby referred as autistic), commonly experience challenges with executive functioning ([4]; [11]; [26]; [58]). Executive function (EF) is important for goal-directed behavior, abstract thinking, social regulation, and decision-making ([10]). These difficulties are associated with significant challenges in daily functioning, particularly in tasks requiring autonomy, problem-solving, and goal setting ([30]). Challenges with executive functioning can also negatively impact autistic individuals’ ability to engage in and develop meaningful social relationships ([7]) and interfere with many life activities ([51]; [62]). One of the factors that is closely tied to EF is sleep ([52]; [57]). Because sleep problems are common in autistic individuals, addressing sleep is important when addressing EF ([52]; [57]). If left unaddressed, EF can lead to poor functional outcomes, and negatively impact overall wellbeing ([12]).

Behavioral interventions such as applied behavioral analysis (ABA) are often recommended for autistic children ([48]). There is substantial evidence that young autistic children can benefit from ABA ([35]; [60]). While ABA can be an effective intervention for young autistic children, autistic individual’s perspectives towards behavioral interventions such as ABA have been overlooked. Criticisms of ABA from autistic individuals include the focus on deficits and conformity to typical behavioral norms or neurotypical behaviors ([1]). Additionally, there is a lack of empirical evidence for recommending ABA to autistic youth and adults who do not have a cooccurring intellectual disability ([5]). The estimated rate of intellectual disability in the autistic population in the United States is just under 40% ([53]). Thus, common ABA models are inappropriate for the majority of autistic youth and adults.

Despite the critical role of EF in supporting autonomy and adaptive behavior, there remains a notable lack of empirically validated EF interventions. Unstuck and On Target (UOT), a school-based program, is one of the few empirically supported EF interventions. In a randomized controlled trial, [30] ([30]) found that students who received UOT demonstrated significant improvements in behavioral flexibility and reductions in executive dysfunction compared to control groups. Outside of UOT, there are some individual EF-focused strategies, such as visual supports, task checklists, and explicit self-regulation techniques that have shown promise for specific aspects of EF ([20]; [46]); however, there is a lack of empirically based EF non-school interventions for autistic youth and adults that address EF more broadly.

An additional barrier for autistic youth and adults seeking an intervention to improve their EF is the lack of neuroaffirming interventions. Unlike many autism interventions which focus on deficits, neuroaffirming interventions recognize that differences are not deficits and that autistic people should be involved in the treatment recommendations that affect them ([34]). Skill acquisition in neuroaffirming interventions focuses on improving the autistic person’s wellbeing and affirming the person’s autistic identity ([24]). While there has been some movement towards neuroaffirming psychological therapy targeting cooccurring conditions such as anxiety or depression ([8]; [18]), there is a dearth of neuroaffirming behavioral interventions for autistic youth and adults. In a study examining predictors of satisfaction with behavioral interventions, autistic adults who had received ABA reported dissatisfaction due to limited opportunities to select personal goals that reflected their own priorities ([39]).

Sleep is closely tied to EF in youth and adults. Lack of sleep and poor-quality sleep (e.g., frequent waking) all negatively affect EF ([52]; [57]). Evidence-based sleep treatments in autistic individuals have primarily focused on behavioral interventions for children. Behavioral strategies such as consistent bedtime routines implemented with active caregiver involvement are particularly effective for younger children ([36]; [45]). While autistic youth and adults also frequently experience significant sleep disturbances ([22]) there is limited research specifically targeting sleep interventions in older autistic youth and adults ([3]).

To address personal goals an individual needs to understand what goals are important to them. Motivational Interviewing (MI) is a collaborative, person-centered approach that helps support individuals in developing their personal goals for behavioral change. When working with both youth and adults, MI emphasizes empathy, respect for autonomy, and the use of open-ended questions, affirmations, reflective listening, and summarizing (OARS; [40]). In the context of autism treatment and neuroaffirming care, MI offers a valuable framework for supporting self-determined goals while honoring individuals’ identities and sensory, cognitive, and emotional differences. Rather than pushing for compliance or externally imposed outcomes, MI helps individuals articulate their own values and motivations, an approach that aligns well with neurodiversity-affirming principles. For autistic youth and adults who may feel misunderstood or pressured to conform in behavioral interventions, MI offers a supportive approach to skill development that is empowering, personally relevant, and aligned with their lived experiences and goals ([44]); thus, it may be a valuable addition to behavioral approaches to interventions.

Manualized interventions have been shown to be effective for autistic people across a variety of differently presenting problems ([2]; [28]; [31]; [33]; [61]). While there are manualized workbooks for executive functioning (e.g., [25]; [50]) and specific manualized workbooks for autistic people covering specific life skills ([19]), less is known regarding the effectiveness of a comprehensive manualized treatment for executive functioning for autistic people that incorporates neuroaffirming interventions such as MI. A neuroaffirming, manualized coaching program that builds executive function (EF) skills—such as time management, organization, and emotional regulation—while also addressing barriers like poor sleep, may enhance engagement and help participants identify personally meaningful reasons for strengthening EF skills.

Given the lack of EF programs for autistic youth and adults and the pervasive influence of executive functioning on developmental trajectories, academic success, meaningful social relationships, employment, and wellbeing, continued investment in the development of EF interventions is essential. There is a strong clinical rationale for advancing structured, evidence-based approaches that support the flexible thinking, self-regulation, and planning skills necessary for success across contexts. Furthermore, there is a need for these interventions to be scalable and accessible to autistic youth and adults. The purpose of this study is to expand the available EF interventions by evaluating a novel manualized neuroaffirming behavioral coaching program, designed to improve executive functioning, social skills, and sleep in autistic youth and adults, and to examine its impact on wellbeing.

## 2. Materials and Methods

### 2.1. Participants and Data Collection

In order to understand the effectiveness of a novel neuroaffirming behavioral coaching program (Compass-Behavioral; Compass-B) for autistic youth and young adults, a convenience sample was utilized. Participant data included in this study was derived from a large non-profit behavioral health organization serving patients in California and Hawaii. Participants were autistic youth and young adults (ages 11–26) who had completed Compass-B as part of their behavioral health treatment and had a confirmed diagnosis of autism spectrum disorder. Autistic participants opted into Compass-B as their care of choice at the onset of behavioral health services.

The organization’s business intelligence department provided archival records of participants who had completed Compass-B upon IRB approval to move forward with this study (IRB protocol 25-065-1352). These records included data pulled from the participant’s medical record in regard to their initial assessment, treatment plan, and discharge reports. Additionally, pre- and post-test data for each module (referred to as Elements) of the manualized program were provided and all demographic information were also included.

### 2.2. Demographic Information

After all data cleaning operations, the sample consisted of 234 autistic participants with an average age at the start of treatment of 15.66 (SD = 2.41). The sample included 98 females (41.9%) and 135 males (57.7%) as well as one non-binary individual (0.4%). For a full description of participant details see Table 1. In addition to the demographic information, further participant information was available, including average length of stay in outpatient treatment (*M* = 4.35 months; SD = 2.70), baseline ABC Vineland score (*M* = 79.38; SD = 10.5), baseline Communication Vineland score (*M* = 83.49; SD = 12.78), baseline daily living skills Vineland score (*M* = 85.40; SD = 13.96), and baseline socialization Vineland score (*M* = 74.43; SD = 13.87). All participants had sufficient language abilities to participate in a coaching program.

### 2.3. Treatment

Compass-B is a manualized program utilizing behavioral principles and Motivational Interviewing, focusing on executive functioning skills and wellbeing within the autistic population. A neuroaffirming framework emphasizing self-determination was an important consideration in the development of Compass-B. Compass-B is conducted in a one-to-one therapeutic format by a licensed mental health clinician or Board-Certified Behavioral Analyst^®^ (BCBA^®^). Compass-B is composed of a self-assessment and seven modules—called Elements (see Table 2) and entails an autistic participant working directly with a trained clinician who acts as a coach. All participants complete the Self-Assessment. Using information from the Self-Assessment, participants work with their coach to choose the Elements that are most important to the participants’ personal goals.

Clinicians who want to conduct Compass-B must be a licensed or certified clinician who have gone through a six-hour Compass-B training. The training includes passing a fidelity check regarding implementation (80% minimum fidelity required to move onto the final exam), as well as a final exam which requires an 80% score to pass. All clinicians who implemented Compass-B for this study met the minimum requirements in order to be a Compass-B coach for autistic youth. The training includes an overview of Compass-B methods and its manual, as well as an in-depth review of ASD, a framework for coaching, and training in Motivational Interviewing. Additionally, all clinicians have mentor sessions with trained Compass-B clinicians. At the conclusion of training, clinicians, referred to as coaches, begin working with participants who have opted into the program.

The length of participation and number of sessions in Compass-B is flexible. Generally, 10 to 14 weekly or bi-weekly sessions over 3–6 months are scheduled depending on participant preference. Typical sessions last between 45 and 60 min. The amount of time a participant spends on an Element is determined by the number of weeks of participation, and the number of Elements chosen is determined by the participant, in partnership with their coach.

At the beginning of the intervention, a self-assessment is conducted which incorporates the participant’s values, strengths, areas of growth, personality traits (e.g., introversion, extraversion), and reviews the seven different Elements. The Self-Assessment, which typically requires approximately four hours, is an important aspect of Compass-B, as it provides participants with the opportunity to learn about themselves and assist their assigned coach in learning about them. Based on all this information, the participant determines which Elements they want to work on and develops personal goals with their coach. Additionally, the Vineland-3 is given during the assessment to understand the participant’s baseline adaptive skills. Once the individual has identified the goals they would like to work on (corresponding to the Elements they choose), they take a pre-assessment to determine their baseline skills in the respective areas.

Each session is structured around the participant’s pre-established personal goals, which were identified during the assessment phase. Session content is drawn directly from the Compass-B manual. The coach facilitates strategy application through interactive methods, including role play, collaborative dialogue, and behavioral skills training (BST) techniques to support the acquisition and generalization of new skills. Some components of the intervention may span multiple sessions, with pacing tailored to align with the participant’s preferences and readiness. Participants may also decide on weekly tasks (e.g., completing a sleep diary; practicing a self-advocacy skill, etc.) to complete between sessions, which are then reviewed collaboratively in subsequent meetings.

Although the intervention follows a structured manual, the approach is person-centered and flexible. Participants play an active role in guiding the direction and focus of sessions to ensure alignment with their goals. Rather than employing a directive model during sessions, the clinician adopts a coaching stance, supporting participants in leading the process and fostering ownership over their growth and outcomes. The coaching model places participants in the lead and encourages them to choose the goals they find valuable.

### 2.4. Instrumentation

#### 2.4.1. Vineland

The Vineland Adaptive Behavior Scale, Third Edition (Vineland-3; [55]) is a norm-referenced assessment evaluating adaptive skills. Adaptive skills are defined as everyday abilities people need to function in their environment. The Vineland includes several domains: socialization, daily living skills, communication, and an optional domain of maladaptive behavior. The Vineland also includes a composite score, the adaptive behavior composite. The Vineland-3 is taken at treatment outset to determine participants’ adaptive functioning.

#### 2.4.2. Pre- and Post-Test Self Assessment

For each Element (or module) of Compass-B, there is a pre-test to determine participants’ baseline skills regarding the skill that Element teaches (e.g., sleep hygiene, setting goals, etc.). At the conclusion of each Element, there is a post-test with the same questions which are used to determine progress in that respective area. For an example of the pre- and post-tests from each Element, see Appendix A.

#### 2.4.3. Self-Observation

The Self-Observation Element includes a pre- and post-test self-assessment that consists of five questions (e.g., “I know how to take basic behavioral data on myself) that is rated on a 5-point Likert scale (e.g., rarely = 1; Sometimes = 2; Not sure = 3; Most time = 4; Always = 5) with a total possible score of 25 points.

#### 2.4.4. Vulnerabilities

The Vulnerabilities Element includes a pre- and post-test self-assessment that consists of six questions (e.g., “I am aware of what makes me vulnerable”) that is rated on a 5-point Likert scale (e.g., rarely = 1; Sometimes = 2; Not sure = 3; Most time = 4; Always = 5) with a total possible score of 30 points.

#### 2.4.5. Self-Advocacy

The Self-Advocacy Element includes a pre- and post-test self-assessment that consists of five questions (e.g., “I know the self-advocacy steps”) that is rated on a 5-point Likert scale (e.g., rarely = 1; Sometimes = 2; Not sure = 3; Most time = 4; Always = 5) with a total possible score of 25 points.

#### 2.4.6. Communicating Socially

The Communicating Socially Element includes a pre- and post-test self-assessment that consists of four questions (e.g., “I have good conversations with people in my life”) that are rated on a 5-point Likert scale (e.g., rarely = 1; Sometimes = 2; Not sure = 3; Most time = 4; Always = 5) with a total possible score of 20 points.

#### 2.4.7. Perspective Taking

The Perspective Taking Element includes a pre- and post-test self-assessment that consists of four questions (e.g., “I understand that others may not feel like I do”) that are rated on a 5-point Likert scale (e.g., rarely = 1; Sometimes = 2; Not sure = 3; Most time = 4; Always = 5) with a total possible score of 20 points

#### 2.4.8. Self-Management

The Self-Management Element includes a pre- and post-test self-assessment that consists of four questions (e.g., “I can set up a plan to improve my self-management skills”) that are rated on a 5-point Likert scale (e.g., rarely = 1; Sometimes = 2; Not sure = 3; Most time = 4; Always = 5) with a total possible score of 20 points.

#### 2.4.9. The Importance of Good Sleep

The Importance of Good Sleep Element includes a pre- and post-test self-assessment that consists of five questions (e.g., ” I get at least 8 h of sleep most nights”) that are rated on a 5-point Likert scale (e.g., rarely = 1; Sometimes = 2; Not sure = 3; Most time = 4; Always = 5) with a total possible score of 25 points.

#### 2.4.10. Catalight Youth Wellbeing Scale

The Catalight Youth Wellbeing Scale (CYWS) is a 15-item scale originally developed to understand the wellbeing of a youth with ASD or I/DD. Each item of the scale has a 5-point Likert scale (1 = Strongly disagree, 2 = Disagree, 3 = Undecided, 4 = Agree, and 5 = Strongly agree) with a maximum score of 75 and a minimum score of 15, with higher scores indicating greater wellbeing. The scale was originally created to have three domains, self-determination, relationships, and self-management, with five items in each domain. Initially, an exploratory factor analysis was conducted, revealing a three-factor structure. Confirmatory factor analysis (CFA) was then used to confirm the three-factor structure. Examination of model fit indices indicated acceptable to excellent model fit, supporting the factor model suggested by the original EFA. However, examination of the AVE suggested that an insufficient amount of variance was explained by the self-management and self-determination subscales. Cronbach’s alpha also indicated that internal consistency for these components was inadequate (<0.70). Model fit indices support the use of three subscales in the youth wellbeing scale, relating to inter/intrapersonal relationships, self-management, and self-determination components. Metrics of validity and reliability are inadequate ([56]).

#### 2.4.11. Exit Survey

At the conclusion of Compass-B, all participants are sent an optional survey to complete, which has three items. Item one states “Compass-B has been helpful for me” and respondents are given a 5-point Likert scale (1 = Not helpful at all; 2 = Not helpful; 3 = Neutral; 4 = Helpful; 5 = Very helpful) to provide a response, with scores ranging from one to five, with higher scores indicating greater helpfulness. Item two asks them to identify which Elements of Compass-B they went through (they check which Elements they completed), and item three asks them to “describe more about how the Elements you picked were helpful to you” in free text.

### 2.5. Data Analysis Plan

To evaluate the effectiveness of Compass-B in a real-world clinical setting, participants completed pre- and post-test measures for each Element completed. A non-experimental group design using paired samples *t*-tests was planned to examine changes from baseline to post-intervention. A priori power analysis conducted in G*Power 3 ([16]) indicated a minimum of 34 participants was required to detect a medium effect size (Cohen’s *d* = 0.5) with 80% power and an alpha level of 0.05.

Because participants completed the Catalight Youth Wellbeing Scale at multiple time points, a repeated-measures ANOVA was planned to assess changes in wellbeing across the intervention. Power analysis suggested that at least 28 participants would be needed to detect a medium effect (Cohen’s *f* = 0.25) with 80% power at an alpha of 0.05, assuming three time points and a correlation of 0.50 among repeated measures ([16]).

Given emerging evidence of sex-related differences among autistic individuals with lower support needs ([9]; [13]; [42]), independent samples *t*-tests were planned to examine baseline differences in adaptive functioning by sex, as measured by the Vineland-3.

All quantitative analysis were conducted in IBM SPSS Statistics for Windows, Version 27 ([27]).

To complement quantitative findings, qualitative free-text responses from the exit survey were analyzed using thematic analysis ([6]) with MAXQDA software ([59]). Two authors experienced in qualitative coding (V.B. & M.B.) independently reviewed and coded the data, with themes being identified through an iterative consensus process.

## 3. Results

### 3.1. Element Pre- and Post-Assessment

A series of paired-samples *t*-tests was run to determine if there were statistically significant mean differences from pre-assessment to post-assessment in each of the Elements included in Compass-B. The results of the analyses revealed significant improvement from pre-assessment to post-assessment on six of the seven Elements. The average number of Elements chosen by participants was 2.19 (range 1–5). The Element chosen most often was Self-Management; the Element chosen least often was Sleep. See Table 3 for the full breakdown of these results. The n in Table 3 represents the number of participants who completed each Element and completed both pre- and post-assessments.

### 3.2. Wellbeing Scale

A one-way repeated measures ANOVA was conducted to determine whether there was a statistically significant difference over the course of three time points (baseline, midway, discharge) of Compass-B. The assumption of sphericity was violated, as assessed by Mauchly’s test of sphericity, χ^2^(2) = 7.08, *p* = 0.029. Therefore, a Greenhouse–Geisser correction was applied (ε = 0.052). The wellbeing scores did not significantly change over the three time points, *F*(1, 29) = 2.41, *p* = 0.131. However, there was mean improvement over time (baseline = 46.90; midway = 53.23, discharge = 54.47).

### 3.3. Sex Differences

Of clinical interest, an independent samples *t*-test was conducted to determine if participant characteristics at baseline were different between the male and female participants. The ABC, daily living skills, and socialization domains were equivalent amongst the male and female participants (*p* > 0.05); however, there was a statistically significant difference regarding communication scores at baseline, with female participants having a significantly higher baseline communication score (*m* = 86.14; SD = 12.93) compared to their male counterparts (*m* = 81.95; SD = 12.58; *p* < 0.05).

### 3.4. Exit Survey and Thematic Analysis

To analyze the exit survey, descriptive statistics and a thematic analysis of the open-ended questions were completed. See Table A2 for a list of all statements. There were a total of 44 respondents to the exit survey. In regard to the question “Compass-B has been helpful for me” the mean score was 4.42, indicating the participants in this program found Compass-B to be helpful to very helpful. Out of the 44 participants who responded to the survey, 23 answered the open-ended question. A sample size of 23 is considered sufficient for thematic analysis to reach thematic saturation ([15]). The methodology described by [6] ([6]) was used for coding, with codes determined by consensus. Multiple codes per answer were allowed to capture the range of themes presented in most of the answers ([6]; [47]). Five themes emerged, each with thematic codes corresponding to the theme: (1) Self-determination, (2) Program Appreciation, (3) Skill gains, (4) Social Understanding, and (5) Wellness Impact. See Table 4 for further details of the thematic map and Table A3 for the count of how often each theme appeared.

## 4. Discussion

Poor executive functioning (EF) abilities in autistic youth and adults have negative consequences for successful day-to-today functioning and long-term wellbeing ([12]). Compass-B is a manualized neuroaffirming intervention utilizing a coaching model for autistic youth and adults that focuses on executive functioning skills and emphasizes self-determination and wellbeing.

### 4.1. Elements Pre- and Post-Assessment

We observed significant improvement in the Compass-B Elements, as measured by pre- and post-testing, for all but one Element. There was a mean improvement based on pre–post-test results for Communicating Socially; however, this did not reach significance. Effect sizes ranged from small–medium to very large. Very large effect sizes were found for Self-Observation and Vulnerabilities, large effect sizes for Self-Advocacy and Sleep, a medium effect size for Perspective Taking, a small to medium effect size for Self-Management, and a small effect size for Communicating Socially.

The Self-Observation Element in Compass-B provides opportunities for participants to learn about themselves and others through observation. The ability to observe one’s own behavior is central to understanding oneself, planning, and recognizing the consequences of personal decisions ([17]). The very large effect size suggests that participants found this Element helpful in understanding themselves and others. As one participant explained, “[Self-Observation] helped me understand my behaviors and other people’s behaviors better.”

Vulnerabilities was the second Element with a very large effect size. Compass-B defines vulnerabilities as “something that affects your emotions and behavior in a negative way.” Understanding and developing a plan for coping with personal vulnerabilities such as not receiving enough sleep or being in a noisy environment can improve an autistic person’s functioning ([41]). Additionally, this Element includes skills for coping with a vulnerability when a vulnerability cannot be avoided (e.g., being in a busy airport). Non-injurious stimming or rhythmic behavior such as rocking can help some autistic individuals cope with vulnerabilities ([37]). Consistent with a neuroaffirming model, the Compass-B Vulnerabilities Element encourages participants to examine the positive role this behavior plays in their lives.

Autistic self-advocacy is an important tenet of the neurodiversity movement. [32] ([32]) suggests that self-advocacy should be included in all interventions intended to support autistic individuals. Despite the importance of self-advocacy skills, many autistic individuals struggle with self-advocacy. [49] ([49]) found that autistic college students rated themselves significantly below their non-autistic peers in their ability to advocate for themselves. The large effect size for Self-Advocacy suggests that participants learned skills that helped them identify when and how to self-advocate. Given the importance of self-advocacy skills, further research should examine the long-term benefits of Compass-B self-advocacy skills.

The Sleep Element, while not directly an EF skill, was added to Compass-B because of sleep’s connection to EF. The Sleep Element was the second Element with a large effect size, suggesting that this Element was effective in improving sleep. While there is evidence that poor sleep has a negative effect on EF abilities in autistic individuals (e.g., [38]) there is little research on how improving sleep can lead to improvements in EF skills. Nevertheless, we found significant improvements in self-reported sleep behavior in participants who chose the Sleep Element. Given the link between EF and sleep, additional research examining the role of improved sleep in EF abilities could expand our understanding of the relationship between sleep, EF, and sleep interventions.

We found a medium effect size for the Perspective Taking Element. Cognitive perspective taking is, in its simplest form, putting yourself in someone else’s shoes. Both feeling understood and understanding another person can improve social relationships ([23]). Understanding someone else’s perspective without necessarily endorsing how the person feels or thinks is one of the skills covered in this Element. This distinction is important, as it allows for understanding without automatically agreeing with someone else. Comments from participants that illustrate this included, “Perspective taking helped with my empathy towards things that I disliked” and “Helped me … have more patience with my sisters.”

The Self-Management Element was the most frequently selected module. Unlike traditional self-management interventions that emphasize behavior reduction (e.g., decreasing externalizing behaviors), the Self-Management Element in Compass-B centers on life management, including strategies for setting priorities and achieving personal goals. The small to medium effect size observed for this Element suggests that participants may have found these skills less useful than those in other Elements, or that the post-test did not adequately capture improvements. Future iterations of Compass-B should consider revising this Element to enhance its effectiveness.

Differences in communication within social situations are a core feature of autism ([54]). The goal of the Communicating Socially Element in Compass-B is not to teach autistic individuals to socialize like neurotypical people, but rather to support their comfort and understanding of social interactions in ways that enhance wellbeing (e.g., “I have good conversations with people in my life”). This Element was the second most frequently selected, indicating strong participant interest. However, the lack of significant change and small effect size suggests that participants may not have experienced the intended benefits. Possible explanations include misalignment between the pre- and post-test measures and actual skill development, or content that does not fully address participants’ needs. Additional research and direct participant feedback are necessary to clarify these findings. As with the Self-Management Element, insights from this study should guide revisions to future versions of Compass-B to strengthen the impact of these Elements.

### 4.2. Wellbeing Scale

Wellbeing is a measure of a person’s overall happiness, satisfaction with life, and positive outlook on life. While we found a mean improvement in wellbeing, this was not significant. Behavioral interventions for autism have historically ignored wellbeing ([29]). More research is needed to clarify the relationship between Compass-B and wellbeing. It is possible that the Elements do not target aspects of wellbeing that are most meaningful to autistic individuals, or that the measure used in this study did not accurately capture participants’ experiences. Given the importance of wellbeing, future research should explore how Compass-B can more effectively address and measure this outcome.

### 4.3. Sex Differences

More males than female participants (one participant identified as nonbinary) enrolled in Compass-B intervention. With the exception of communication, baseline adaptive behavior scores were equivalent between males and females. Female participants had significantly higher baseline communication scores. The higher communication scores for females brought the average adaptive score for females into the adequate range while the average adaptive score for males was in the moderately low range. The communication score difference is consistent with other studies finding that autistic males and females without intellectual disability have, on average, disparate communication skills, with autistic females having greater communication skills. In a study of 2146 autistic children and adolescents, [14] ([14]) found that females had better communication abilities, as measured by the Vineland-II. Similarly, in a longitudinal study, [43] ([43]) reported that, overall, females tended to score higher in communication abilities on the Vineland regardless of neurodivergent traits (e.g., autism and attention deficit hyperactivity disorder).

### 4.4. Thematic Insights

Thematic coding of the exit survey comments resulted in five themes. The most frequent theme was Self-Determination. This theme captures the journey toward becoming an informed, confident, and proactive decision-maker in one’s own life. It reflects the interplay between understanding oneself, striving for personal growth, speaking up for one’s needs and rights, and exercising independence in daily decisions. This theme supports the inclusion of the Compass-B Self-Assessment and Self-Advocacy Element. Growth, Focus, and Self-Awareness were the most frequent codes within the Self-Determination theme. A comment that illustrates the importance of this theme is “It was helpful because it taught me to look back on why I did things and reflect to determine if I wanted to improve or if it was a good thing.”

Highlighting the importance of neuroaffirming social abilities, the second most frequent theme was Social Understanding. Several participants noted that the program enhanced their ability to interpret social cues, understand other perspectives, and maintain healthy boundaries. This theme encompassed both understanding others and fostering empathy (e.g., “… communicating socially helped me understand where my boundaries should be in different situations”). The results of the thematic analysis and the lack of significant improvement in the Communicating Socially Element, coupled with the high frequency of the Communicating Socially Element in participants’ comments suggest that consideration should be given to modifying the Communicating Socially Element to better meet the needs of Compass-B participants.

Wellness Impact was the third most frequently endorsed theme. Participants described positive changes to their overall wellbeing, such as better sleep habits, improved mood, and improved daily functioning (e.g., “Setting goals around sleep and getting exercise helped me be in a better mood”). Wellness gains were often linked to specific program elements like Sleep and Self-Management.

The Skills Gain theme described acquiring or strengthening practical skills (e.g., “I learned good coping skills from the vulnerability element”). This included better coping, improved time management, and the ability to set and achieve personal organizational goals. The emphasis was on tangible, applicable abilities that could be integrated into daily routines.

Program Appreciation was the least endorsed theme. This theme reflected the value participants placed on the program’s flexibility, relevance, and personalization. Being able to choose the Elements underscored the importance of self-determination (e.g., “…helpful because I got to pick subjects [Elements] I needed to improve on and then work throughout the months on bettering myself with my [coach].”

Overall, the thematic analysis indicated that self-determination is a significant strength of Compass-B, with participants citing growth focus, self-awareness, self-advocacy, and autonomy as key gains. The choice and flexibility in the program’s design were valued and contributed to engagement and perceived relevance. Skill gains in coping, time management, and organization directly support independence and improved social interactions. Wellness outcomes add an important dimension to the program’s impact, supporting sustainable change beyond skill acquisition.

### 4.5. Limitations

While this study had a number of strengths, including a large sample size and emphasis on neuroaffirming practices, there are also some noteworthy limitations. Most importantly, there is the lack of a control group and random assignment. The absence of a control group means it is not possible to compare Compass-B to another intervention or no intervention at all, which could lead to causal conclusions. Future research comparing Compass-B to a control condition could provide further evidence for this coaching program. In order to learn more about the effectiveness of Compass-B, funding for a randomly controlled trial is being considered. Additionally, except for the Vineland-3, which was used only for baseline comparison, we did not use any standardized measures. While the pre- and post-tests measured the participant’s self-report of how beneficial an Element was, a standard measure of EF skills such as the Behavior Rating Inventory of Executive Functioning, Second Edition (BRIEF-2; [21]) would have provided additional information. It is also possible that the pre- and post-test do not measure the EF skills they were intended to measure. Additional research using a standardized EF measure (e.g., BRIEF-2) along with the Element pre- and post-tests would strengthen our understanding of how Compass-B affects EF skills. Future research should include a standardized measure of EF.

## 5. Conclusions

This study provides initial support for Compass-B. Incorporating what has been learned in this study into future versions of Compass-B and collecting additional outcome data are important next steps in the development of Compass-B. In addition to adding a standardized EF outcome measure, it will be important to use participant and coach feedback. We will use participant and coach feedback to improve Element content. Finally, given the short-term nature of Compass-B, collecting long-term data to examine how well Compass-B skills generalize over time is important.

## Figures and Tables

**Table 1 behavsci-15-01667-t001:** Ethnicity.

Ethnicity	N	%
Asian	39	17%
Black/African American	4	1.7%
Caucasian/White	75	32%
Declined to state	33	14%
Hispanic/Latino	28	12%
Multi-Ethnic	49	21%
Other	6	2.3%

**Table 2 behavsci-15-01667-t002:** Elements of Compass-Behavioral.

Element	Description
Self-Assessment	All participants complete the assessment. This includes a review of the participants’ strengths, values, and what they want to focus on in Compass-B.
Self-Observation	In this module, participants practice non-judgmental self-observation by mapping what happens before and after their behavior to better understand patterns and support meaningful change.
Vulnerabilities	In this module, participants explore how common vulnerabilities—often experienced simultaneously—can influence their ability to learn, solve problems, and respond to situations.
Self-Advocacy	In this module, participants learn the what, why, and how of becoming their own advocate, even when it feels difficult to speak up or communicate clearly.
Communicating Socially	In this module, participants learn about different types of relationships and strategies for improving communication within each.
Perspective Taking	In this module, participants learn perspective taking—considering situations from another person’s point of view—to better understand others and their actions.
Self-Management	In this module, participants learn self-management skills to organize their time, manage their thoughts and feelings, and take steps toward achieving their personal goals.
The Importance of Good Sleep	In this module, participants learn how sleep affects health and wellbeing, explore the impact of too little or poor-quality sleep, and discover ways to improve their sleep.

**Table 3 behavsci-15-01667-t003:** Paired-samples *t*-test.

Element	*n*	Pre-Assessment		Post-Assessment		*t*	*p*	*Cohen’s d*
		M	SD	M	SD			
Communicating Socially	31	12.06	3.7	13.97	5.2	−1.83	0.07	0.33
Perspective Taking	19	10.00	5.3	13.05	5.6	−2.21	0.04	0.51
Self-Advocacy	18	8.06	4.8	15.06	4.4	−4.14	<0.001	0.98
Self-Management	39	12.44	3.8	15.36	5.0	−2.67	0.011	0.43
Self-Observation	19	10.21	5.3	18.47	5.2	−4.68	<0.001	1.07
Sleep	12	12.67	4.50	18.58	5.9	−2.68	0.021	0.77
Vulnerabilities	20	16.50	6.8	23.95	6.5	−7.06	<0.001	1.58

**Table 4 behavsci-15-01667-t004:** Thematic analysis.

Theme	Self Determination	Program Appreciation	Skills Gains	Social Understanding	Wellness Impact
Subthemes	Growth Focus	Choice and relevance	Coping Strategies	Communication skills	Sleep
	Self-Awareness		Time Management	Empathy and listening	Wellbeing
	Self-Advocacy		Organizational skills		
	Autonomy				

## Data Availability

The data that support the findings of this study are derived from participant medical and treatment records at a large non-profit behavioral health organization and contain sensitive health information. Due to privacy and ethical restrictions, these data are not publicly available. De-identified data may be made available from the corresponding author upon reasonable request and with coordination from the agency’s Office of Risk Management.

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
