# Peer review of "Initial Analysis of the Effectiveness of Compass-Behavioral for Autistic Youth: A Community-Based Retrospective Analysis"

_behavsci, 2025, doi:10.3390/bs15121667_

Round 1
Reviewer 1 Report
Comments and Suggestions for Authors
Thank you for the opportunity to review this manuscript. Overall, the authors present an interesting synthesis of a novel behavioral coaching intervention. Including data on the acceptability of this intervention is a strength of this work. However, more information is needed to understand the purpose of creating this intervention, who completed specific assessments and how they are interpreted, and how to contextualize these findings for future research.
Introduction:
The introduction is well-written and summarizes key points. However, it seems to jump between multiple topics, and it is unclear how each of these topics tie together. For instance, the authors are encouraged to more clearly link the paragraphs on sleep, neurodiversity-affirming intervention, and motivational interviewing to the aims of their intervention summarized in this manuscript.
Please revise this sentence for clarity: Beyond UOT, there are some individual EF-focused strategies, such as visual supports, task checklists, and explicit self-regulation techniques have shown promise for specific aspects of EF (Gilotty et al., 2002; Reichow & Volkmar, 2010). Additionally, how does Compass-B build on UOT? Why did the authors decide to create their own intervention instead of using UOT? Please address.
The authors are encouraged to add a “current study” paragraph to the end of the introduction that clearly outlines 1) what this intervention adds relative to prior literature, 2) the professionals who created Compass-B and the manual, and 3) the theoretical framework for this intervention and primary outcomes.
Methods/ Results:
The authors refer to “average length of stay.” Is this an inpatient or outpatient program? Please clarify in text.
Please add the language/IQ level of participants. It sounds like an intervention for verbally fluent youth.
Please elaborate on what the therapist’s “coaching stance” entails and explain how therapists met criteria for implementing the intervention with fidelity.
Authors are encouraged to add the psychometric properties for the Catalight Youth Wellbeing Scale.
The sample sizes in Table 3 do not add up to the total sample size included. Please explain who completed each pre- to post- assessment and how to interpret these findings in the discussion.
Discussion
The sleep element could be a whole standalone intervention. Please explain how each intervention component works together and whether there is sufficient intervention time to address the wide range of Elements. If not, this should be addressed in the discussion/limitations.
The authors are encouraged to add where they plan to go next with this line of research and whether they hope that this will become a manualized intervention that will be evaluated in RCTs.
Author Response
See file attached.

Reviewer 2 Report
Comments and Suggestions for Authors
Dear Authors,
Thank you for the opportunity to review your work. I found it very interesting. I believe the information you provide is highly relevant.
The study planning, data collection, and analysis procedures are appropriate. Using a combination of quantitative and qualitative analysis techniques is appropriate. The repeated measures design was key in assessing the relevance and rigor of this work.
I have made some recommendations that primarily affect the report's presentation format.

Round 2
Reviewer 1 Report
Comments and Suggestions for Authors
The authors have addressed my comments and the paper is now appropriate for publication.